# Research on a New Type of Rigid-Flexible Coupling 3-DOF Micro-Positioning Platform

**DOI:** 10.3390/mi11111015

**Published:** 2020-11-18

**Authors:** Guilian Wang, Yong Wang, Bingrui Lv, Ruopeng Ma, Li Liu

**Affiliations:** 1Tianjin Key Laboratory for Advanced Mechatronic System Design and Intelligent Control, School of Mechanical Engineering, Tianjin University of Technology, Tianjin 300384, China; wangguilian@tjut.edu.cn (G.W.); 15022292499@163.com (Y.W.); lvbingrui@tju.edu.cn (B.L.); Maruopengsky@163.com (R.M.); 2National Demonstration Center for Experimental Mechanical and Electrical Engineering Education (Tianjin University of Technology), Tianjin 300384, China

**Keywords:** micro-positioning platform, rigid-flexible coupling, three degrees of freedom, flexible hinge

## Abstract

A new type of rigid-flexible coupling three degrees of freedom (3-DOF) micro-positioning platform with high positioning accuracy and high bearing capacity is developed, which consists of flexible drive mechanism and rigid platform. The flexible drive mechanism consists of three sets of symmetrical parallel round flexible hinge structures, each with a wedge structure in the middle of the symmetrical parallel flexible hinge. The rigid platform has an inclined plane with the same angle as the wedge, while the wedge structure is used to achieve the self-locking effect. The flexibility matrix method and ANSYS are used to analyze the statics of the flexible drive mechanism. The first four natural frequencies of the platform are obtained by dynamic simulation analysis. A symmetrical rigid flexible coupling micro-positioning platform experimental system is developed. Output characteristics, positioning accuracy, relationship between frequency and amplitude, and bearing performance of the micro-positioning platform are tested. These experimental results obviously show that the micro-positioning platform has good motion characteristics, high positioning accuracy, large movement distance, and large load bearing capacity performance.

## 1. Introduction

Flexure hinge has the advantages of convenient processing, low cost, easy realization of miniaturization and high positioning accuracy. It is often used in ultra-precision instruments such as micro motion device, micro-positioning device, precision positioning stage in atomic force microscope, optical automatic focusing system and optical fiber docking device. Flexible hinge mobile platform has important application value in micro nano manufacturing, micro electro mechanical system (MEMS), scanning electron microscope, micro gripper, manipulator, even 3D printing, robot and other precision engineering fields [1,2,3,4,5,6] due to its high efficiency, no backlash transmission, high resolution and precise control, simple and compact structure [7,8].

The micro-positioning platform based on flexure hinge can provide the required output displacement. Compared with the traditional linkage mechanism, flexure hinge has the advantages of no friction, no gap, repeatable movement, no lubrication and low cost [9,10,11]. The flexure hinge is used as a rotatory/translational joint with elastic deformation, and is used for precise motion of micro-positioning platform. In order to make the flexure hinge have higher motion accuracy, the great stiffness in the non-functional direction must be kept, so as to ensure the non-parasitic motion as small as possible to satisfy the motion accuracy. Many scholars have studied it. Polit et al. [12] designed a high bandwidth and high precision flexure hinge mechanism for nano manufacturing. Li et al. [13] developed a flexible hinge motion platform with laminated structure based on the limited operators of electron microscope. Lv et al. [14] designed a novel three degrees of freedom (3-DOF) motion device based on flexure hinge, which can realize closed-loop control of large output stiffness. Kim et al. [15] designed a fast tool servo mechanism based on flexure hinge, which is mainly used for diamond turning microstructure manufacturing. Because of these extensive applications, the flexible hinge 3D mobile platform has important research value in its operation range, operation accuracy, natural frequency, resolution, decoupling, degree of freedom, stability, anti-interference ability and other aspects [16,17]. In the current research on the flexible micro-positioning platform, it mainly includes the exploration of the operating range, operating accuracy, natural frequency, and degree of freedom. The application of flexure hinge mechanism in micro-nano manufacturing, electron microscope and other precision instruments requires the movement device to have a larger range of travel [18,19]. For displacement amplification, there are several types such as lever type, bridge type, and Scott-Russell mechanism. Kim et al. [20] studied a self-guided displacement magnification mechanism with bridge magnification. Wang et al. [21] proposed a 3-DOF nano-positioning platform with three two-level lever amplification and achieved a high-precision positioning work space, whose dimension is 283.13 μm × 284.78 μm × 8.73 mrad. Zhang et al. [22] designed a 3-DOF parallel micro-motion platform using a bridge-type and lever-type composite amplifier. The experimental results show that the output displacement of the positioning platform is more than 30 times the input displacement of the piezoelectric actuator. Zhu et al. [23] designed a two-stage displacement-amplified flexure hinge mechanism based on the Scott–Russell half-bridge magnification mechanism. Experiments show that the magnification rates of movement in the X-direction and Y-direction are 5.2 and 5.4, respectively. Tian et al. [24] designed a 3-DOF nano-positioning platform based on an L-shaped lever and a half-bridge magnifying structure. The experiment showed that the actual working range of the positioning platform output is 128.1 μm × 131.3 μm × 17.9 μm. The coupling error in the non-working direction is less than 1.56% and the motion resolution is 8 nm.

For other indicators, such as decoupling, compact structure, high stability, multi degree of freedom and other aspects, many scholars also have done more in-depth research. Lee et al. [25] designed a 2-DOF motion device. In order to solve the decoupling problem, a closed-loop feedback method was adopted, with a bandwidth of 100 Hz. Guo et al. [26] designed a compact 3-DOF micro/nano positioning platform based on the plane control principle of three piezoelectric actuators. In order to make the flexure hinge mechanism more compact, Li et al. [27] proposed the design of Z-type hinge. Cai et al. [28] designed a flexible motion platform for a precision positioning control system with 6-DOF by using three symmetrical T-shaped hinges and three elliptical flexure hinges. Chen et al. [29] studied a large-scale remote motion mechanism with input/output decoupling. The experiment shows that the maximum coupling motion is less than 1%. Li et al. [30] use three pairs of parallel four-bar design of flexure hinge mechanism of a 3-DOF platform for micro-positioning platform. 

At present, for the development and application of micro-positioning platform, many scholars have done a lot of research, designed a variety of micro/nano positioning platform, and achieved a lot of research results. However, there are also some shortcomings, including the following points: (1) The structure is complex, it is difficult to work in a high bandwidth. (2) The motion of each axis is difficult to obtain the same dynamic performance. (3) The force on the output end of the micro-positioning platform is borne by the flexure hinge mechanism, which makes the positioning platform not suitable for the working conditions with large bearing capacity changes. In order to make the micro-positioning platform have larger bearing capacity and higher positioning accuracy, a new rigid flexible coupling 3-DOF micro-positioning platform is designed by using parallel circular flexure hinge as micro-positioning mechanism and piezoelectric ceramic actuator. The input-output relationship of the micro-positioning platform is determined by using the theoretical analysis and finite element simulation analysis of the micro-positioning platform. The experimental platform is built. Besides, the motion characteristics, positioning accuracy, and bearing capacity of the micro-positioning platform are tested.

## 2. Mechanism Design and Analysis

### 2.1. Mechanism Design

A new type of rigid-flexible coupling 3-DOF micro-positioning platform is designed. The structure of this platform is relatively simple, and it is easy to analyze and process. The overall structure is shown in Figure 1. The platform is mainly composed of two parts: a flexible drive mechanism and a rigid platform. The flexible drive mechanism is composed of three groups of symmetrical parallel circular flexure hinge structures, which have the small structural parasitic movement, strong load-bearing capacity and small dynamic bad coupling. Meanwhile, a wedge structure is set in the middle of each group of symmetrical parallel flexure hinges, and the wedge angle is 45°. The rigid platform has an inclined plane with the same angle as the wedge for easy placement.

The movement of this traditional flexure hinge device is realized by the deformation of the flexure hinge, which has advantages of frictionless drive and strong anti-interference ability. Tang et al. [31] designed a micro-positioning platform with fast response and maximum crosstalk less than 1.2%. Liu et al. [32] designed a three-axis precision positioning device using a low-interference motion piezoelectric actuator. However, the flexible structure must have greater flexibility in two decomposition directions at the same time when this type of device achieves force decomposition through a flexible hinge. For example, when Z-shaped hinge structure is used as force decomposition unit, it is required to have greater flexibility in both decomposition directions. Therefore, when the external force is added, it is easy to cause hinge deformation and produce disturbance error [33]. The micro-positioning platform proposed in this study is composed of two parts, namely flexible drive mechanism and a rigid platform. The flexible drive mechanism and the rigid platform are connected by inclined surface contact to form a rigid kinematic pair. The rigid kinematic pair can not only play the role of force decomposition, but also ensure that the device has a large output rigidity, which is beneficial to improve the anti-interference ability and the load-bearing capacity of the device. Meanwhile, the piezoelectric ceramic driver is protected by the reverse friction self-locking feature, which is especially suitable for precise positioning of large-quality workpieces.

### 2.2. Flexibility Matrix Method Modeling

The flexible driving mechanism of the rigid flexible coupling 3-DOF micro-positioning platform is mainly composed of symmetrical parallel circular flexure hinges. The structure has strong bearing capacity, which can effectively reduce the parasitic motion of the mechanism and ensure the stable output of the mechanism. Since the three groups of flexure hinge mechanisms are the same and have a left-right symmetrical structure, only one of the flexure hinge mechanisms needs to be analyzed during the analysis, as shown in Figure 2. The specific parameters are shown in Table 1.

In this paper, the flexible hinge mechanism of the micro-positioning platform is modeled with full flexibility by using the matrix-based compliance modelling (MCM) method. The mechanism can be regarded as A_1_ and A_2_ in parallel, and then connected with A_3_ block in series. As shown in Figure 3a, the moving platform part is calculated as right angle flexure hinge. In Figure 3b, $P_{i}-x_{i}y_{i}z_{i}\;$is the Cartesian coordinate system at the i-th hinge connecting arm $P_{i}$, and $P_{i}-x_{i}y_{i}z_{i}$ is the local flute of the i-th flexible hinge on the i-th flexible chain Carr coordinate system.

The flexibility matrix of A_1_ output to $O_{1}^{\text{’}}$ can be defined as:(1)$$C_{A_{1}}^{O_{A}}=C_{O_{1}}^{O_{A}}+C_{P_{1}}^{O_{A}}+C_{O_{1^{\prime }}}^{O_{A}}=T_{O_{1}}^{O_{A}}C_{i}^{R}{\left(T_{O_{1}}^{O_{A}}\right)}^{T}+T_{P_{1}}^{O_{A}}C_{i}^{j}{\left(T_{P_{1}}^{O_{A}}\right)}^{T}+T_{O_{1^{\prime }}}^{O_{A}}C_{i}^{R}{\left(T_{O_{1^{\prime }}}^{O_{A}}\right)}^{T}$$

The flexibility matrix of A_2_ output to $O_{2}^{\text{’}}$ can be defined as:(2)$$C_{A_{2}}^{O_{A}}=C_{O_{2}}^{O_{A}}+C_{P_{2}}^{O_{A}}+C_{O_{2^{\prime }}}^{O_{A}}=T_{O_{2}}^{O_{A}}C_{i}^{R}{\left(T_{O_{2}}^{O_{A}}\right)}^{T}+T_{P_{2}}^{O_{A}}C_{i}^{j}{\left(T_{P_{2}}^{O_{A}}\right)}^{T}+T_{O_{2^{\prime }}}^{O_{A}}C_{i}^{R}{\left(T_{O_{2^{\prime }}}^{O_{A}}\right)}^{T}$$

A_1_ is connected with block A_2_ in parallel and block A_3_ is connected in series. The flexibility matrix of motion platform to O_A_ point can be expressed as follows:(3)$$C_{L}^{O_{A}}={\left[{\left(C_{A_{1}}^{O_{A}}\right)}^{-1}+{\left(C_{A_{2}}^{O_{A}}\right)}^{-1}\right]}^{-1}+T_{P_{3}}^{O_{A}}C_{i}^{L}{\left(T_{P_{3}}^{O_{A}}\right)}^{T}$$
where $C_{A_{1}}^{O_{A}},C_{O_{1^{\prime }}}^{O_{A}},C_{A_{2}}^{O_{A}},C_{O_{2^{\prime }}}^{O_{A}}$ represents the flexibility matrix of flexure hinge $C_{i}^{C}\left(i=1,1^{\prime },2,2^{\prime }\right)$ transformed into $O_{A}-X_{A}Y_{A}Z_{A}$ coordinate system. $C_{P_{1}}^{O_{A}},C_{P_{2}}^{O_{A}},C_{P_{3}}^{O_{A}}$ represents the flexibility matrix of flexure hinge $P_{i}^{C}\left(i=1,2,3\right)$ transformed into $O_{A}-X_{A}Y_{A}Z_{A}$ coordinate system. $T_{O_{1}}^{O_{A}},T_{O_{1^{\prime }}}^{O_{A}},T_{O_{2}}^{O_{A}},T_{O_{2^{\prime }}\;}^{O_{A}}$ represents the flexibility matrix of flexure hinge $O_{i}^{C}\left(i=1,1^{\prime },2,2^{\prime }\right)$ transformed into $O_{A}-X_{A}Y_{A}Z_{A}$ coordinate system. $T_{P_{1}}^{O_{A}},T_{P_{2}}^{O_{A}},T_{P_{3}}^{O_{A}}$ represents the flexibility matrix of flexure hinge $P_{i}^{L}\left(i=1,2,3\right)$ transformed into $O_{A}-X_{A}Y_{A}Z_{A}$ coordinate system.

Since the drive flexible hinge mechanism of the micro-positioning platform has symmetry, the overall flexibility of the mechanism is:(4)$$C^{O_{A}}={\left[{\left(C_{L}^{O_{A}}\right)}^{-1}+\left(T_{\mathrm{ry}}\left(\phi \right)C_{L}^{O_{A}}{\left(T_{\mathrm{ry}}\left(\phi \right)\right)}^{T}\right)\right]}^{-1}$$
where $T_{\mathrm{ry}}\left(\phi \right)$ is the rotation matrix with respect to the y-axis rotation angle.

### 2.3. Simulation Analysis

#### 2.3.1. Static Analysis

In order to verify the correctness of the flexibility model of the flexible drive mechanism in the micro-positioning platform, this paper uses ANSYS to conduct a static simulation analysis of the flexible drive mechanism. Since the mechanism is composed of three identical flexible driving mechanisms, only one flexible driving mechanism needs to be analyzed in the static analysis. In the finite element simulation, the solid element with 10 nodes and 187 element type is selected. The material is aluminum alloy, whose modulus is 70 GPa and Poisson’s ratio is 0.3. The equivalent load of 100 N is applied to the input end of the piezoceramic No. 1 shown in Figure 4. Meanwhile, full displacement constraints are applied to all three ends of the positioning platform mechanism to detect the deformation characteristics of the flexible hinge mechanism. The finite element analysis results and the MCM model analytical calculation results are shown in Table 2.

According to the analysis in the Table 2, the displacement variation error calculated by comparing the two methods is 7.06%. This indicates that the micro-positioning platform has accuracy modeling and small error, which can effectively model the design structure and meet the design requirements. There are some reasons accounting for the errors: (1) The mesh generation of finite element is not precise, and there are inevitable calculation errors. (2) When MCM method is used, the deformation of symmetrical over constrained mechanism is not considered as nonlinear deformation, which leads to inaccurate modeling accuracy. (3) The modeling method of typical flexure hinge is not perfect.

#### 2.3.2. Modal Analysis

In order to avoid the damage of resonance to the platform, ANSYS is used to simulate the flexible drive mechanism, and the natural frequency of the corresponding vibration mode is obtained. Entity cell 10 node 187 cell type is selected. The material is aluminum alloy, the elastic modulus is 70 GPa, Poisson’s ratio is 0.3, and the density is 2700 Kg/m^3^. The first four order modal nephogram of the flexible driving mechanism is obtained as shown in Figure 5. The first four natural frequencies are 4508.21 Hz, 4590.59 Hz, 4890.59 Hz, and 6020.79 Hz, respectively. The first-order natural frequency is relatively high, and the first three-order modes resonate along the two motion axes respectively. They are significantly different from the resonance frequency of the fourth-order mode (above 2000 Hz), which is much larger than the frequency in actual work, indicating that the flexible driving mechanism of the micro-positioning platform has ideal dynamic characteristics and can be applied in practical work.

## 3. Experiment

### 3.1. Experimental Method

As shown in Figure 6, The computer sends the electrical signal to the Piezo amplifier modules through the control software, and the Piezo amplifier modules and transmits the electrical signal to the piezoelectric ceramic. Piezoelectric ceramics are excited by electrical signals to generate driving force, which acts on the flexible hinge mechanism of the micro-positioning platform to generate displacements in the X, Y and Z directions. Meanwhile, they are detected by the capacitive sensor and fed back to the computer to obtain the displacement curve. Figure 7 shows the micro-positioning platform test system, which is mainly composed of a modular piezo controller, a P-842.60 piezoelectric ceramic driver, a D-100.00 capacitive displacement sensor, a capacitive sensor mount, a computer, an air-floating vibration isolation table, Level instrument and so on. The micro-positioning platform is fixed on the air-floating vibration isolation table. The capacitive displacement sensor, the piezoelectric ceramic driver and the computer are respectively connected with the power amplifier, and the capacitive displacement sensor is connected with the rigid platform through the capacitive sensor fixing frame. Figure 8 shows the installation of capacitive sensors along the X, Y and Z output directions. It is driven by the No. 2 piezoelectric ceramic in the X-direction, it is driven by the No. 1 piezoelectric ceramic in the Y-direction. It is driven by the No. 1, 2, and 3 piezoelectric ceramics in the Z-direction.

### 3.2. Results and Analysis

#### 3.2.1. Output Characteristic 

In order to better test the output characteristics of the micro positioning platform, this paper selects two groups of different amplitude sinusoidal signals, a group of triangular signals, a group of complex signals and a group of step signals as the output response test content. The sinusoidal voltage signal with period of 0.08 s, amplitude of 40 V and 90 V is used as the input signal. The output trajectory curves of the micro-positioning platform in X, Y and Z directions are detected by capacitance sensor, and the trajectory tracking performance curves in X, Y and Z directions are obtained, as shown in Figure 9. 

Figure 9 shows the trajectory following performance of sinusoidal response signal, where Figure 10a–c are the trajectory tracking performance curves in X, Y and Z directions, respectively. As can be seen from Figure 9, the micro-positioning platform has good sine signal tracking performance. When the input period is 0.08 s and the amplitude of sinusoidal voltage signal is 40 V, the output trajectory of the micro-positioning platform presents a sinusoidal regular change. The experimental results show that the trajectory tracking performance of the micro-positioning platform will be significantly improved when the vibration amplitude of the input sine signal is small.

Using the triangle wave signal with period of 0.32 s and amplitude of 0 to 90 V, the output trajectory curves of the micro-positioning platform in X, Y and Z directions are detected by capacitance sensor, and the trajectory tracking performance curve in the Figure 10 of the micro-positioning platform in each direction is obtained. For complex signals, the three-way output trajectory curve of the micro-positioning platform is shown in Figure 11. The three-way output response curves of step signal are shown in Figure 12. The blue curve is the theoretical trajectory, and the red curve is the experimental trajectory.

Figure 10 shows the trajectory following performance of triangular wave signal, where Figure 10a–c are the trajectory tracking performance curves in X, Y and Z directions, respectively. When the input period is 0.32 s and the amplitude changes from 0 V to 90 V, the output trajectory of the micro-positioning platform in X, Y and Z directions shows regular changes, but there are obvious differences between the theoretical trajectory and the experimental trajectory in the cusp part. Due to the response speed of the piezoelectric actuator, the material rebound speed of the device and the structure of the device, the output trajectory of the micro-positioning platform presents a smooth arc trajectory in the cusp tribe. 

Figure 11 shows the complex signal trajectory tracking performance, where Figure 11a is trajectory tracking performance curve in the X-direction, Figure 11b is trajectory tracking performance curve in the Y-direction, and Figure 11c is trajectory tracking performance curve in the Z-direction. The input complex signal is synthesized by sine wave and triangle wave. The micro-positioning platform has better tracking performance for complex signal trajectories in three directions, but the theoretical trajectory and the experimental trajectory are obviously different, and the device output response is immediately distorted when the input signal is converted from a sine wave to a triangle wave. 

Figure 12 shows the trajectory tracking performance of step signal, where Figure 12a is the trajectory tracking performance curve in the X-direction, Figure 12b is the trajectory tracking performance curve in the Y-direction, and Figure 12c is the trajectory tracking performance curve in the Z-direction. When the input signal suddenly drops from 40 V to 0 V, it takes about 0.09 s for the output of the micro-positioning platform to return to the steady state in X, Y and Z directions, which indicates that the output response of the micro-positioning platform to the sudden change of signal has hysteresis phenomenon. The vibration amplitude in the Z-direction is larger than that in X and Y directions because the wedge in the Z-direction is not rigidly connected with the rigid platform.

#### 3.2.2. Positioning Accuracy

The precision test system of 3-DOF micro-positioning platform is shown in Figure 7. The electrical signal of the system is directly edited by computer and transmitted to piezoelectric actuator through power amplifier. The electrical signals are set to make the micro-positioning platform generate small square wave displacement square wave signals in X, Y and Z directions, and the capacitance sensor is used to detect the three-way output signals. The output precision test curves of X, Y and Z directions are obtained as shown in Figure 13. By observing the curve in Figure 13, it is found that the displacement change of the platform is 12 nm in the X-direction, 7 nm in the Y-direction, and 60 nm in the Z-direction, respectively. Because the wedge is not rigidly connected with the rigid platform in the Z-direction, the accuracy in the Z-direction is obviously lower than that in the X-direction and Y-direction, resulting in a large displacement error.

#### 3.2.3. Frequency and Amplitude

In order to test the dynamic performance of the platform, a sine wave excitation voltage with an initial voltage of 90 V and a period of 0.32 s is given. Under the same voltage of 90 V, the sine wave excitation signals with frequencies of 5 Hz, 10Hz, 20 Hz, 25 Hz, and 50 Hz are set, and the output trajectory changes measured in three directions are recorded by capacitive sensor. The observed output displacement curves are all standard sine wave graphs. The amplitude variation diagram of frequency and output trajectory under different frequencies is drawn, as shown in Figure 14. The output trajectory curve at 10 Hz is shown in Figure 15.

In Figure 14, when the input signal is 5 Hz, the maximum displacement of the micro-positioning platform is 32.4 μm in the X-direction, 25.5 μm in the Y-direction, and 40.2 μm in the Z-direction. When the frequency changes from 5 Hz to 50 Hz, the track amplitude of X, Y and Z directions is more and more close, and it can be seen that the higher the frequency, the more consistent the amplitude in the three directions (as the frequency increases, the faster the speed, the greater the damping (damping is proportional to the speed), so the amplitude becomes smaller). As can be seen from Figure 15, the micro-positioning platform has the largest travel in the Z-direction, and the change trend of trajectory curve in three directions is relatively smooth and stable.

#### 3.2.4. Bearing Performance

The wedge structure designed in this paper is mainly to increase the bearing capacity and stability of the micro-positioning platform, and avoid the influence of large force on the normal operation of the micro-positioning platform. The test method of bearing performance is shown in Figure 16, which is composed of computer, modular piezo controller, capacitance sensor and micro-positioning platform. The computer is connected with the capacitance sensor through the power amplifier, and the capacitance sensor is connected to the output of the piezoelectric ceramic driver through auxiliary parts. The test system of bearing performance is shown in Figure 17. In order to test the bearing performance and stability of the micro-positioning platform, a load from 1 N to 10 N is applied at the center of the rigid platform. The displacement variation of the flexure hinge driving mechanism is measured with a capacitive sensor, and the average value of each load three times is recorded, and the force displacement diagram is drawn, as shown in Figure 18. The displacement curve of 5 N load is shown in Figure 19.

It can be seen from Figure 19 that the curve is relatively gentle when no-load is applied, and the displacement change is 0.739 μm when load is 5 N. It can be seen from Figure 18 that the output stiffness of the flexure hinge driving mechanism is 12.304 N/μm, and the displacement variation of the flexure hinge driving mechanism presents a good linearity when the weight on the rigid platform is increasing. When bearing the above load, the force of flexure hinge driving mechanism on the piezoelectric actuator is small. The results show that the micro-positioning platform can bear the above load and has good bearing performance.

## 4. Conclusions

(1)A symmetrical rigid-flexible coupling micro-positioning platform is designed, which is composed of three groups of parallel and symmetrical flexible drive mechanisms and rigid platforms. Each group of symmetrical parallel flexible hinges is provided with a wedge structure in the middle.(2)The flexibility matrix method is used to model the flexible driving structure of micro-positioning platform. When equivalent load (100 N) is applied to the output end of a group of piezoelectric ceramic actuators, the deformation of flexure hinge mechanism is 47.8 μm. the static simulation analysis with ANSYS shows that the deformation is 52.4 μm, and the error is 7.08%. The first-order natural frequency of the flexible driving mechanism is 1797.03 Hz, which shows that the micro-positioning platform has a high natural frequency.(3)A symmetrical rigid flexible coupling micro-positioning platform was developed. The output characteristics, positioning accuracy, the relationship between frequency and displacement amplitude and the bearing capacity of the micro-positioning platform were tested. The positioning accuracy and output displacement response have good tracking performance to the given initial voltage, which shows that the platform has high positioning accuracy and good motion characteristics. The micro-positioning platform has a large stroke and can work in a low frequency state. When the frequency increases from 5 Hz to 50 Hz, the amplitude of the output trajectory of the micro-positioning platform decreases with the increase of the frequency, and the travel in the Z-direction is greater than that in the X-direction and Y-direction. Through the bearing performance test of the micro-positioning platform, the output stiffness of the flexure hinge driving mechanism is 12.304 N/μm, and the force on the piezoelectric ceramic actuator is small when the micro-positioning platform is under load, which indicates that the micro-positioning platform has good load-bearing performance.

## Figures and Tables

**Figure 1 micromachines-11-01015-f001:**
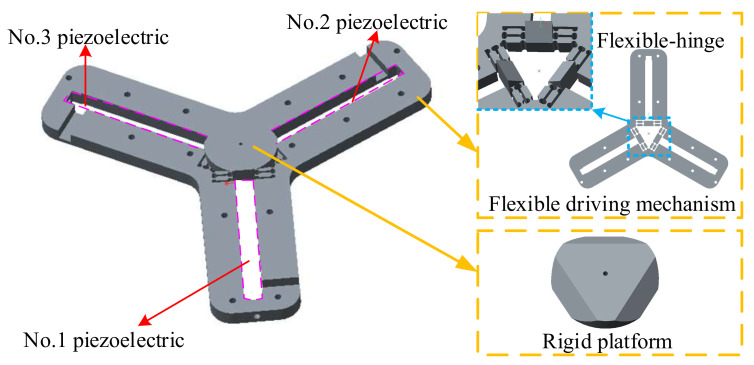
Symmetrical rigid-flexible coupling micro-positioning platform.

**Figure 2 micromachines-11-01015-f002:**
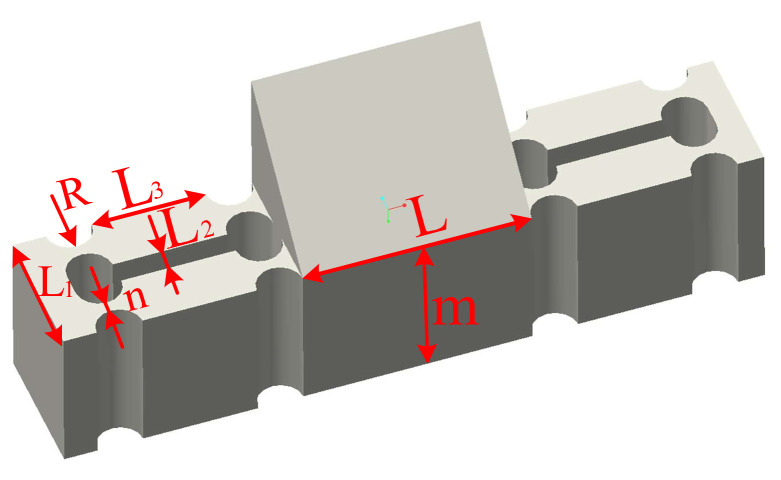
Drive flexure hinge model.

**Figure 3 micromachines-11-01015-f003:**
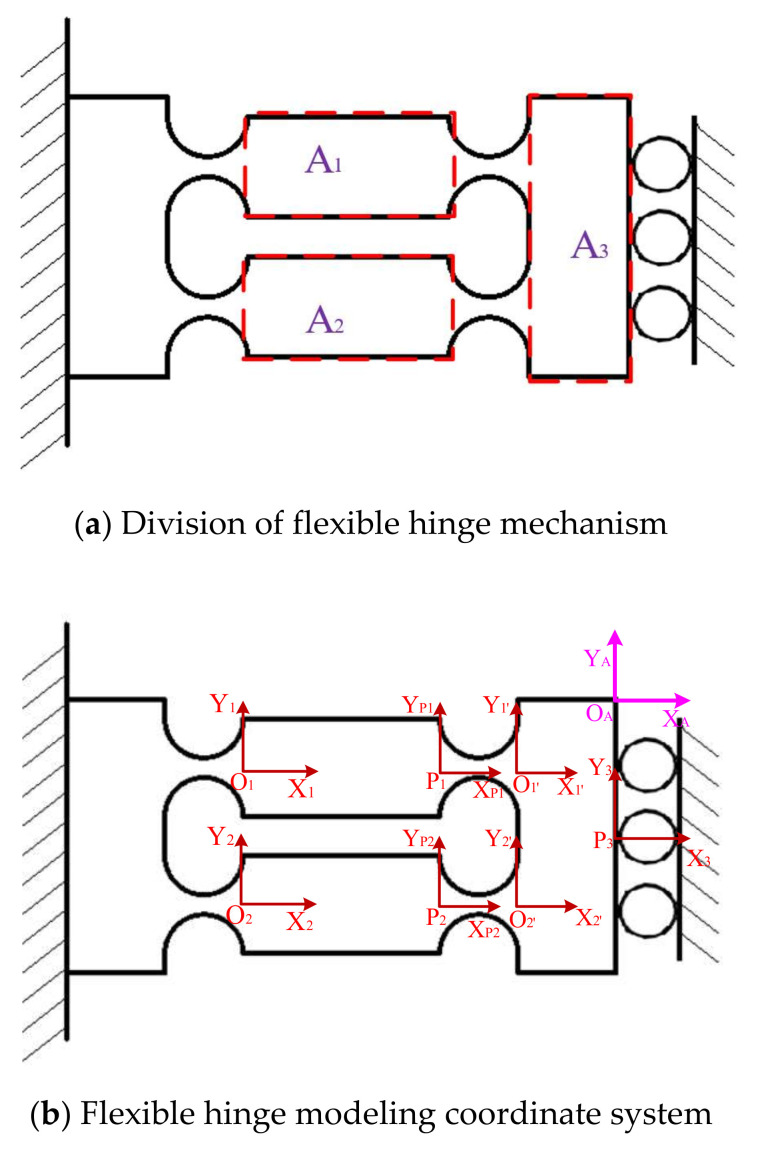
MCM model of driving flexible hinge.

**Figure 4 micromachines-11-01015-f004:**
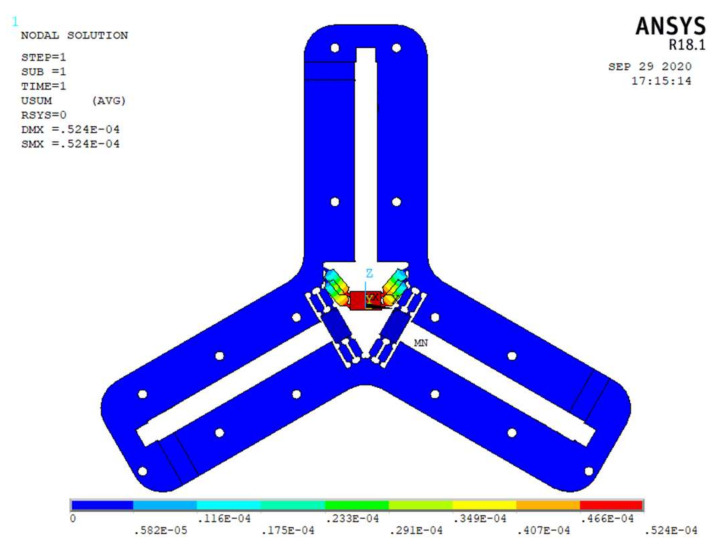
Displacement in the Y-direction.

**Figure 5 micromachines-11-01015-f005:**
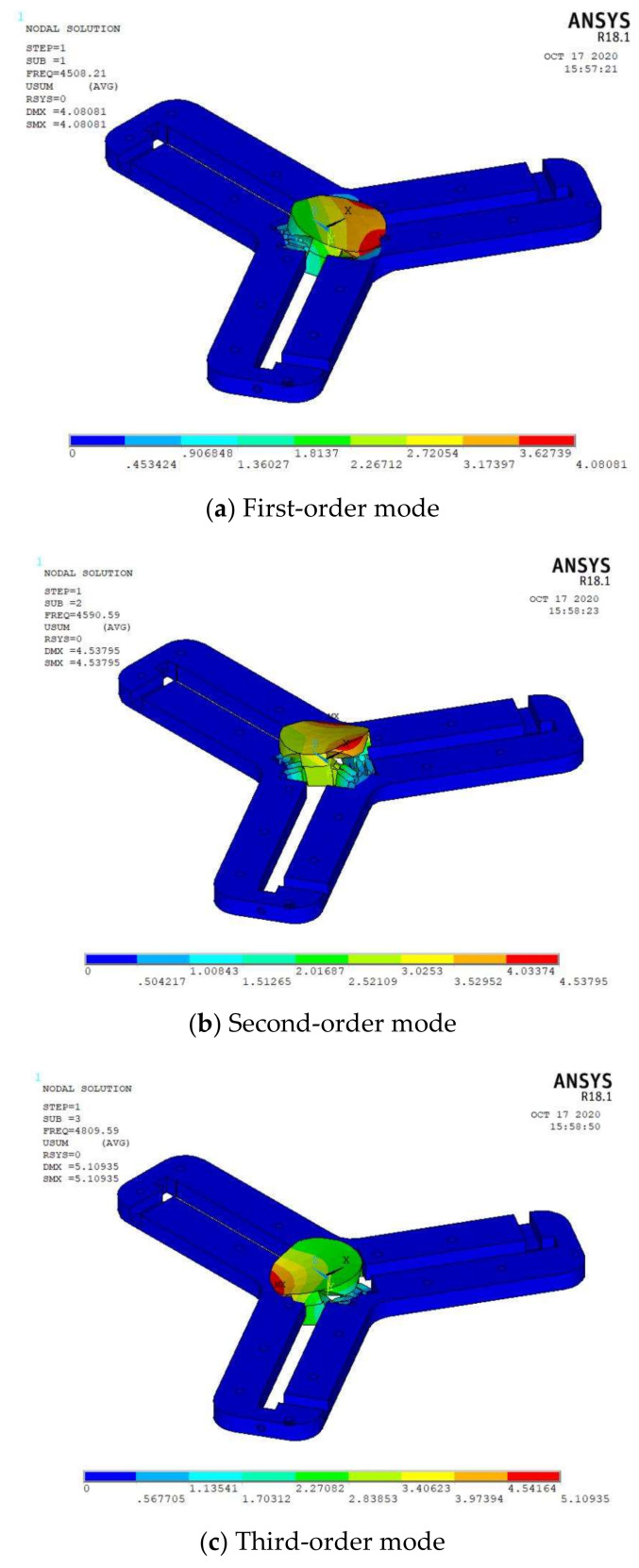
Modal analysis.

**Figure 6 micromachines-11-01015-f006:**
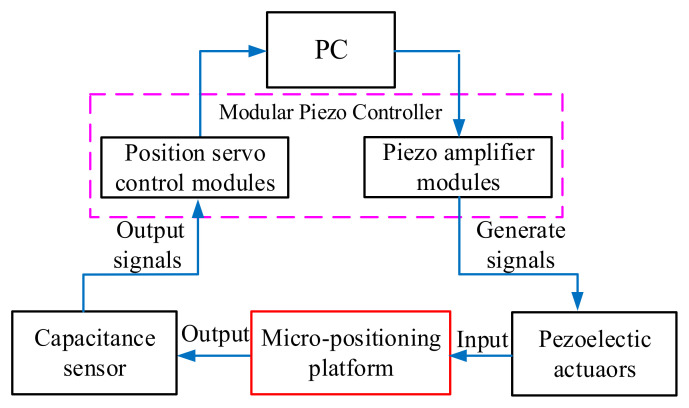
Experimental test method.

**Figure 7 micromachines-11-01015-f007:**
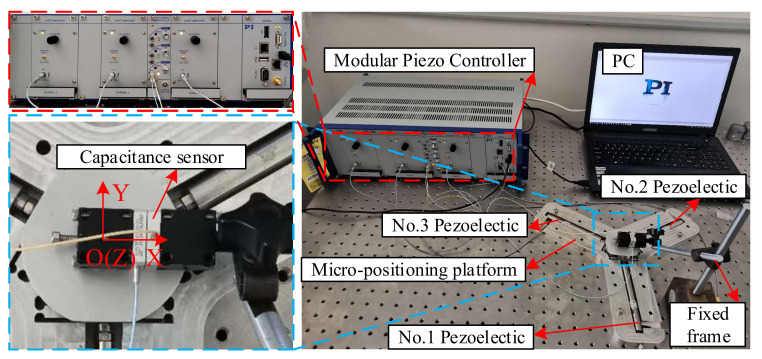
Micro-positioning platform test system.

**Figure 8 micromachines-11-01015-f008:**
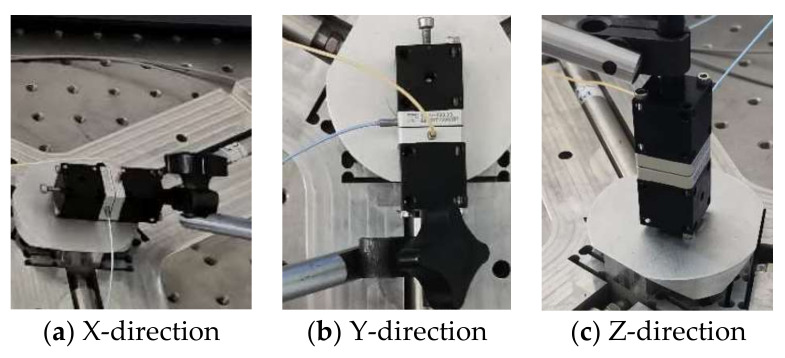
Capacitive sensor installation in all directions.

**Figure 9 micromachines-11-01015-f009:**
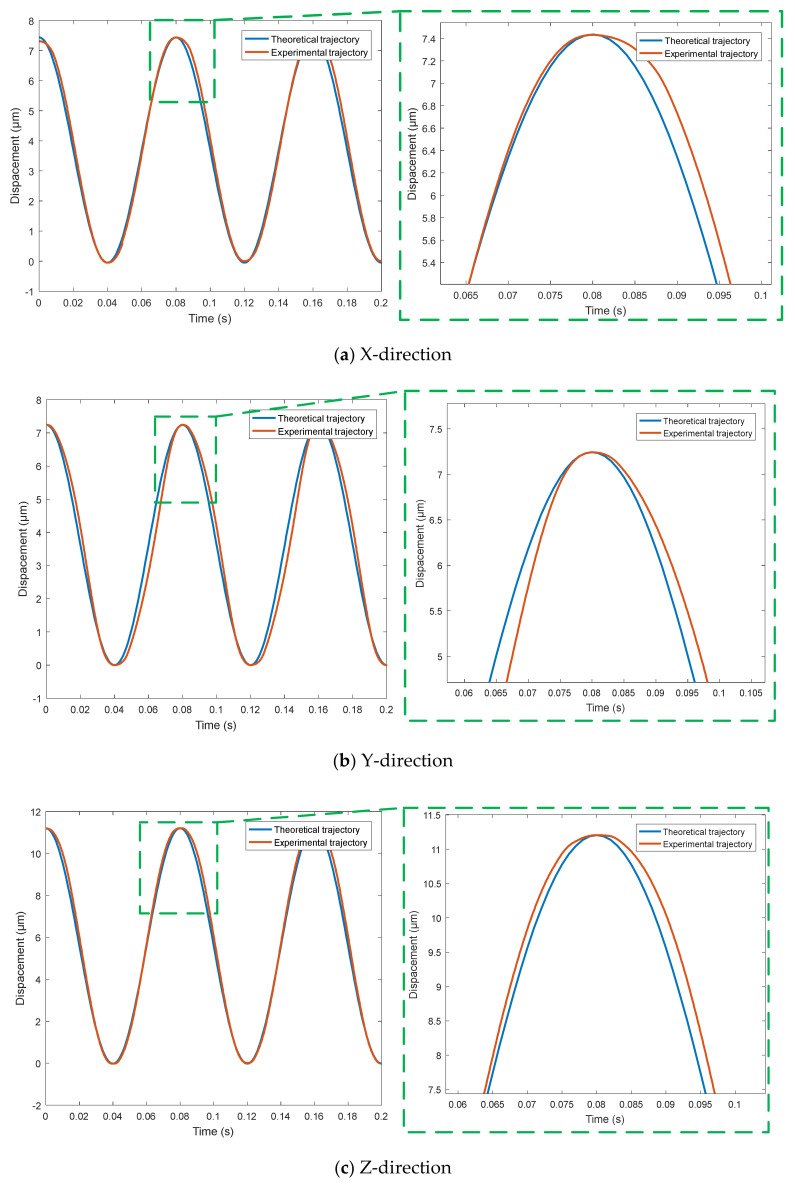
Trajectory tracking performance of sinusoidal response signal.

**Figure 10 micromachines-11-01015-f010:**
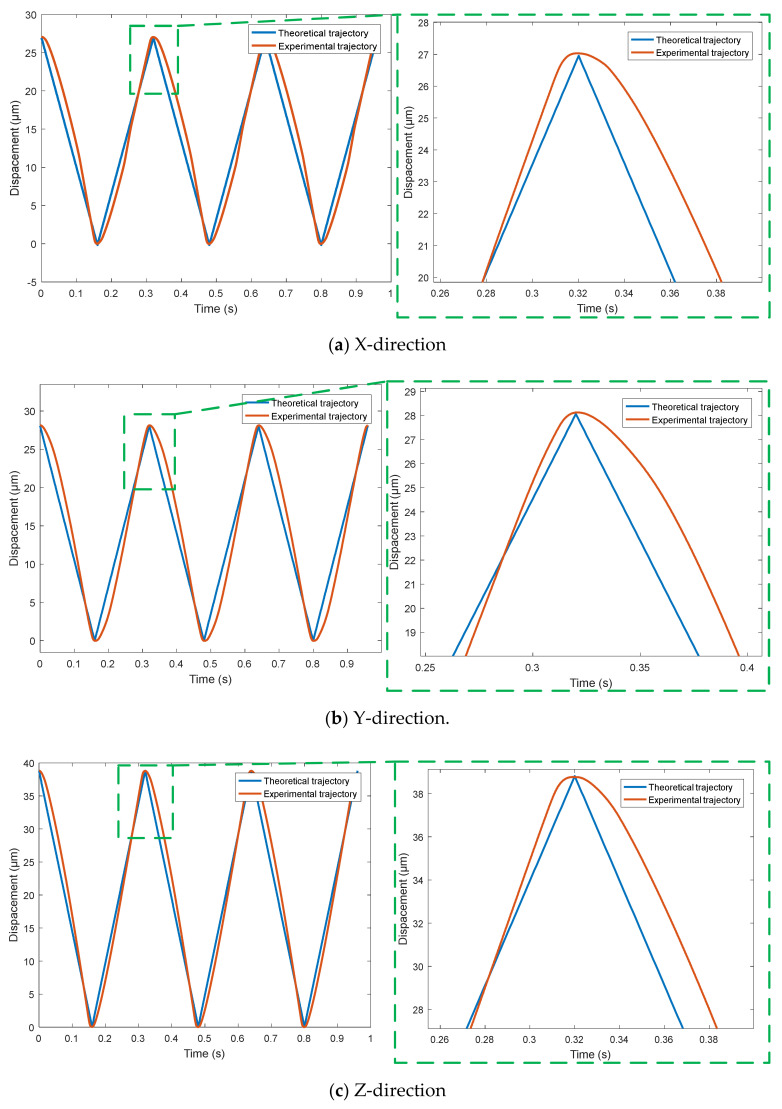
Trajectory tracking performance of triangular wave signal.

**Figure 11 micromachines-11-01015-f011:**
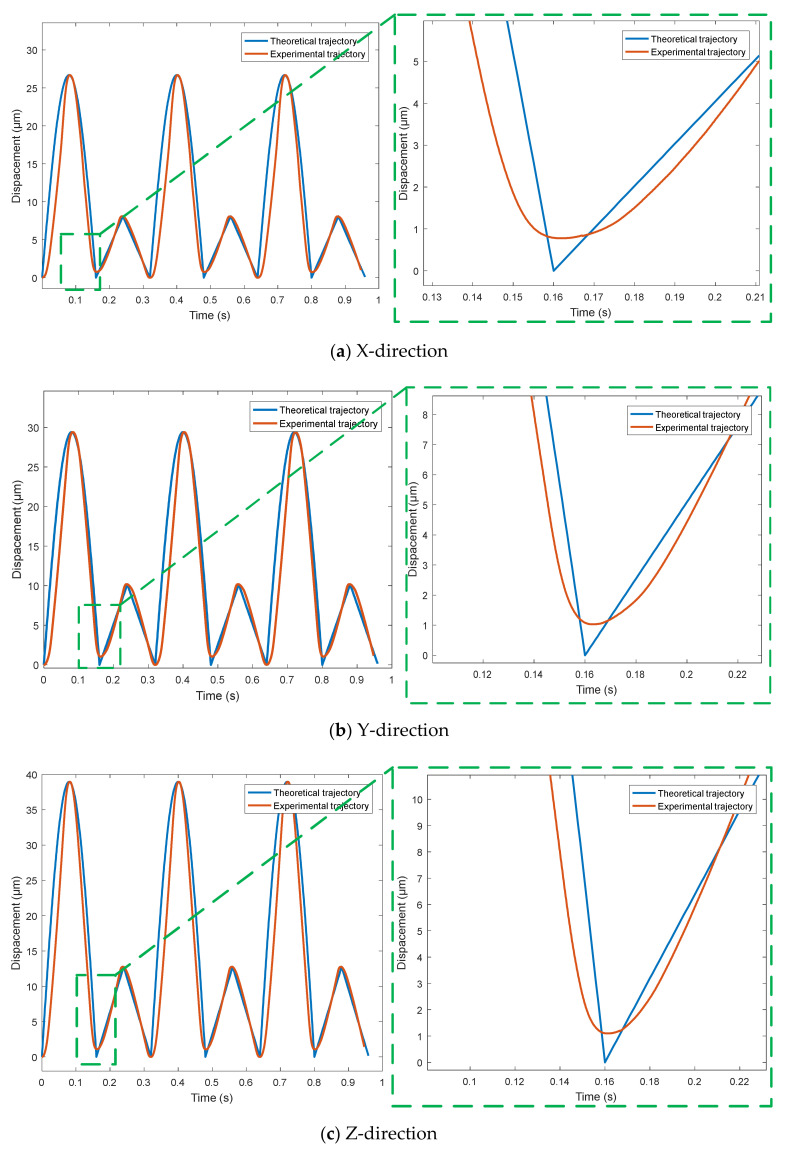
Trajectory tracking performance of complex signal.

**Figure 12 micromachines-11-01015-f012:**
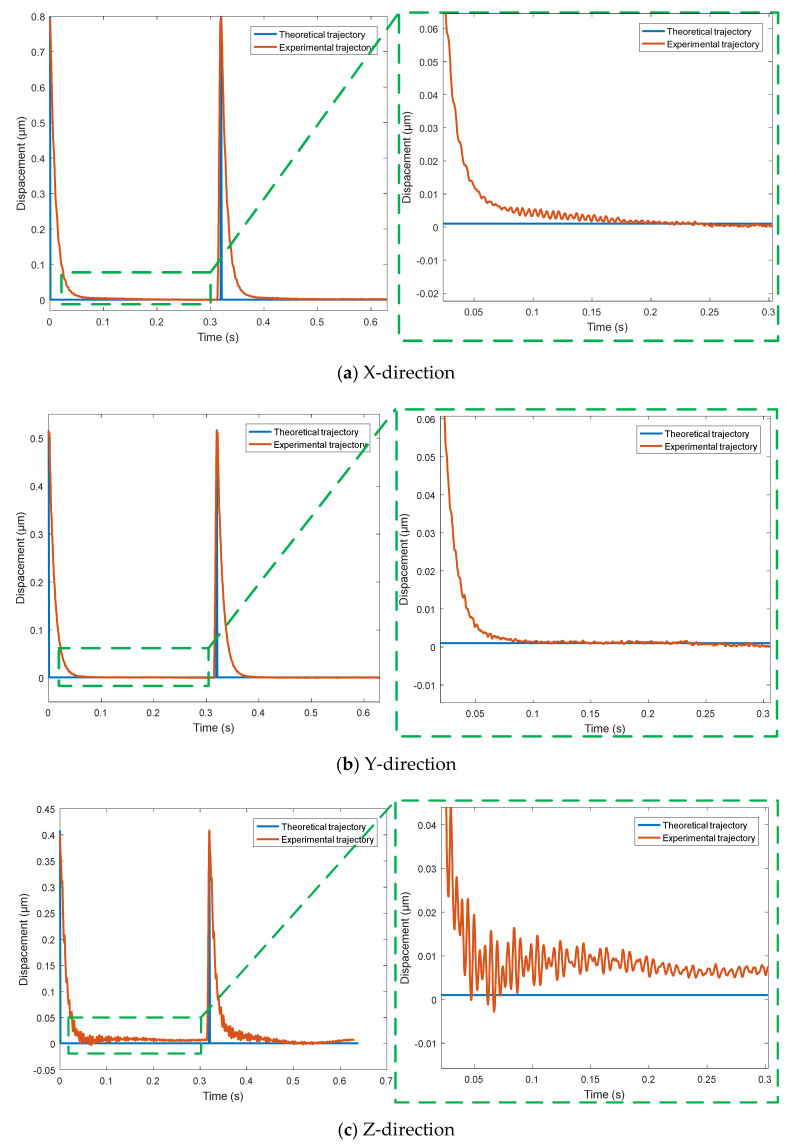
Trajectory tracking performance of step signal.

**Figure 13 micromachines-11-01015-f013:**
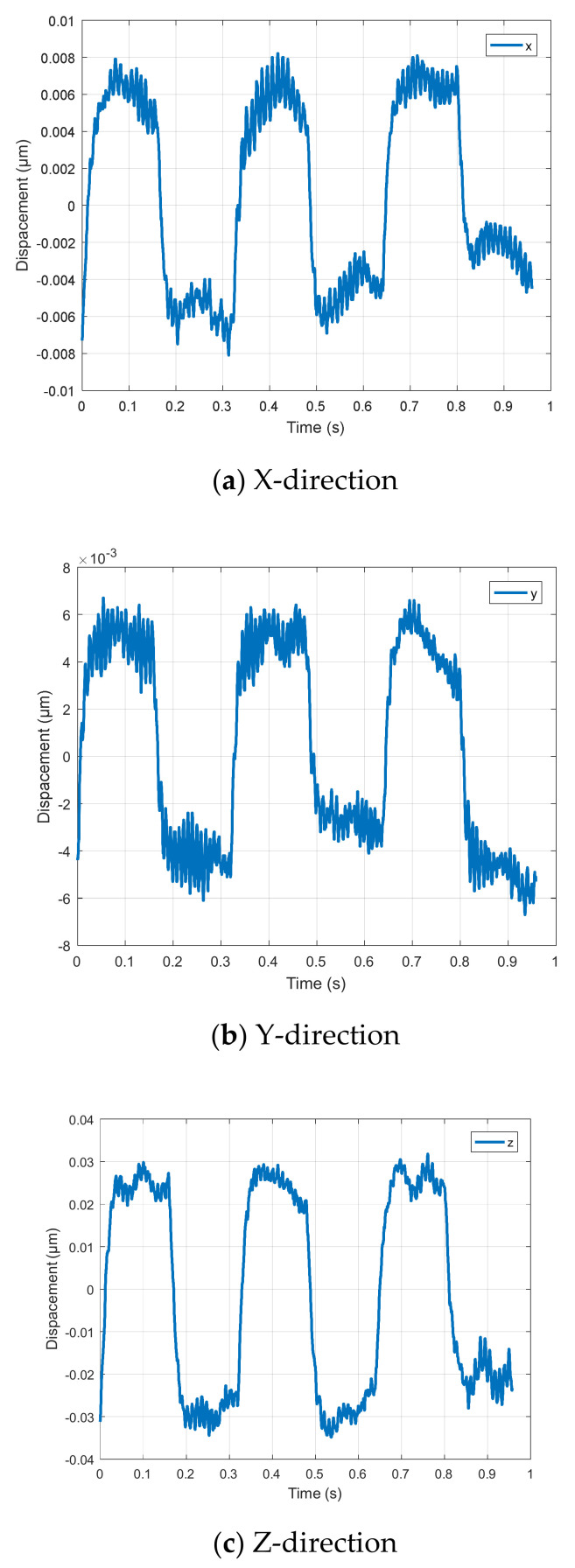
Precision test curve.

**Figure 14 micromachines-11-01015-f014:**
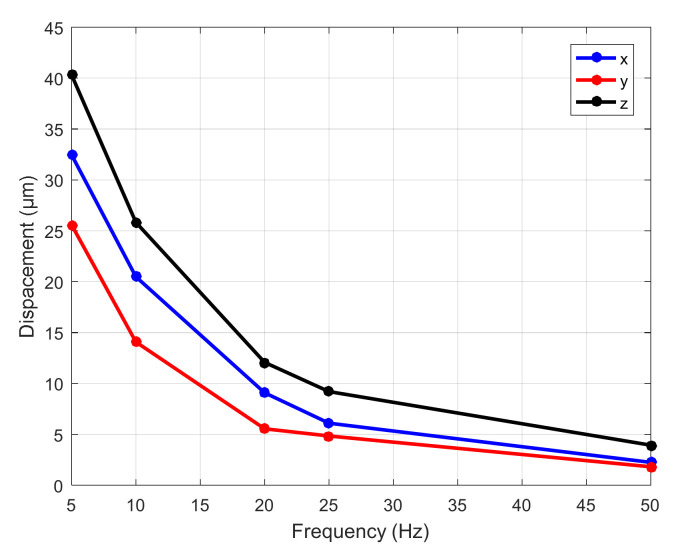
Variation of frequency and output trajectory amplitude.

**Figure 15 micromachines-11-01015-f015:**
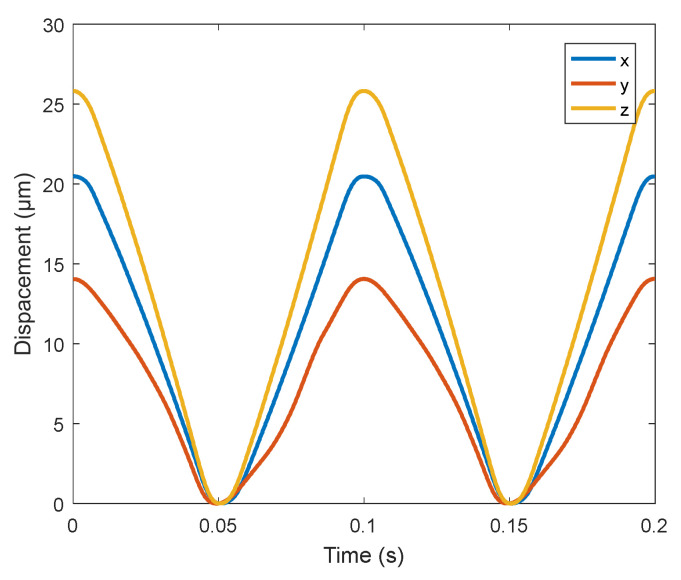
Output trajectory curve at 10Hz.

**Figure 16 micromachines-11-01015-f016:**
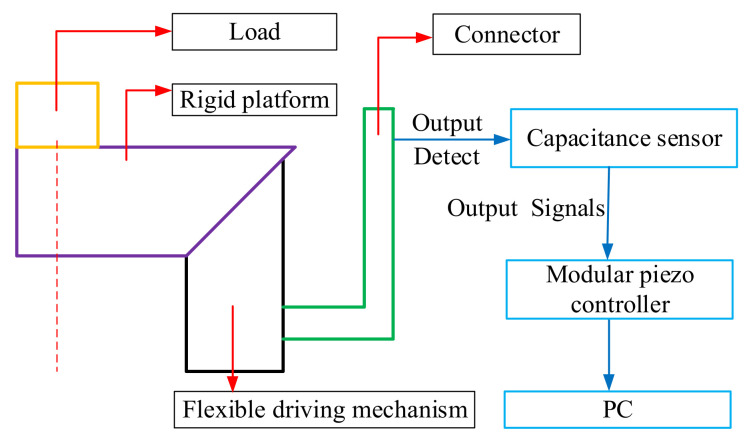
Test method of bearing performance.

**Figure 17 micromachines-11-01015-f017:**
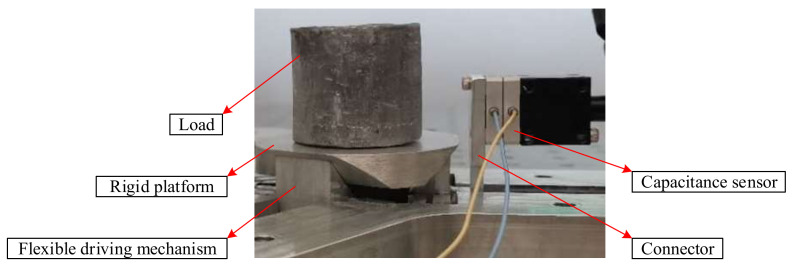
Bearing performance test.

**Figure 18 micromachines-11-01015-f018:**
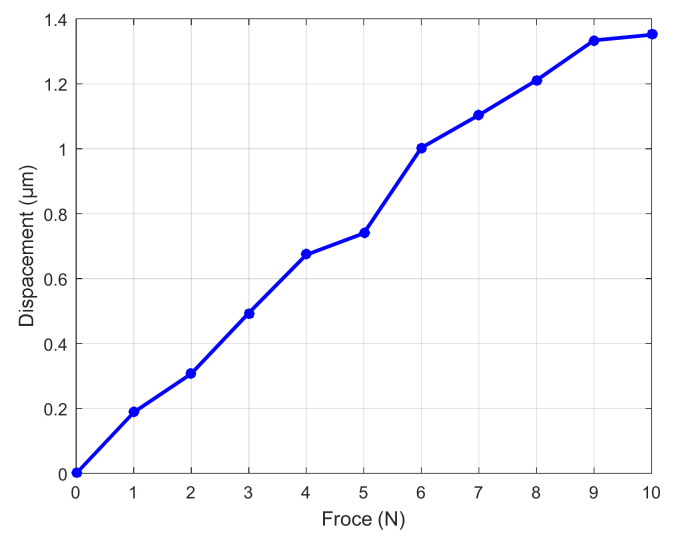
Variation of load-displacement.

**Figure 19 micromachines-11-01015-f019:**
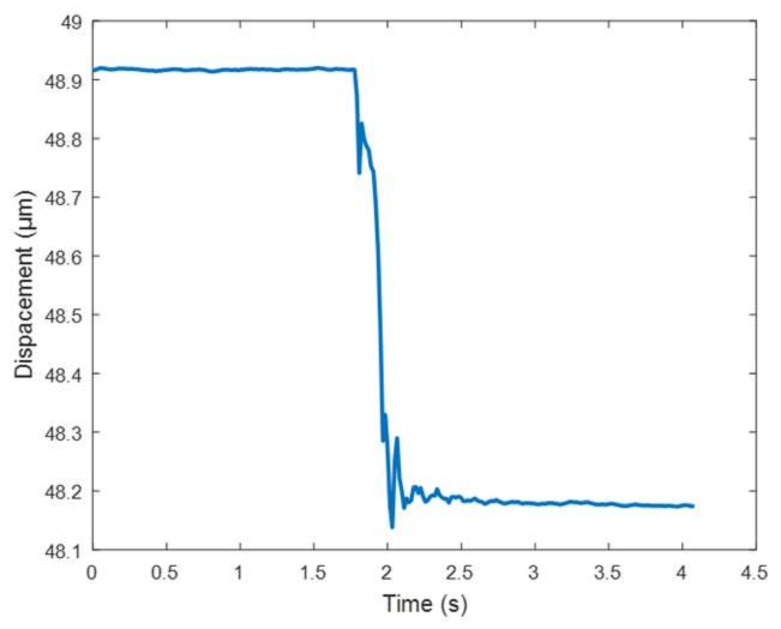
Displacement change at 5 N.

**Table 1 micromachines-11-01015-t001:** Specific parameters of mechanism.

L(mm)	L_1_(mm)	L_2_(mm)	L_3_(mm)	R(mm)	m(mm)	n(mm)
20	12	2	10	2	14	1

**Table 2 micromachines-11-01015-t002:** Comparison of finite element analysis method and MCM method.

	Force(N)	Variation of Displacement(μm)	Error
Finite element	100	52.4	7.06%
MCM	100	48.7	

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
