# Peer review of "Research on a New Type of Rigid-Flexible Coupling 3-DOF Micro-Positioning Platform"

_micromachines, 2020, doi:10.3390/mi11111015_

Round 1
Reviewer 1 Report
This paper develops an interesting new type of rigid-flexible coupling three degrees of freedom (3-DOF) micro-positioning platform with high positioning accuracy and high bearing capacity. Also some related experiments are conducted. For consideration of publication, I would like to suggest the authors to consider the following comments:
1. There are 'flexible drive mechanism' and 'flexible driving mechanism' used within the whole paper. It will be better if there is only one consistent expression used.
2. In section 2.3.2 Dynamic analysis, there is only modal analysis, so it should be changed as Modal analysis.
3. In section 2.3.2, the lowest natural frequency is about 1797 Hz without considering the 'rigid platform' part. And in section 3.2.3, the frequency test shows that with 50 Hz frequency, the amplitude changes. As this test is conducted with the rigid platform(I assume), so in section 2.3.2, it will be better to include the results of modal analysis with the rigid platform.
4. From figure 10-13, when the amplitude is 40V, the displacement is 0-7,0-11 in figure 10; when the amplitude is 90V, the displacement is 25-50, 80-110, 65-11 in figure 11; The same issues also happen in figure 12-13. The authors should carefully check the corresponding displacements under different amplitudes of voltages and then modify these figures.
5. It would be better if the authors could summarize the characteristics of the developed micro-positioning platform in a table and add also some comparison with some other similar platform already developed in other references.
6. There is some recent literature which the authors can include in the references to make the paper cover more recent development of micro-positioning platform, for example:
Li, P.-Z., etc. Piezoelectric Actuated Phase Shifter Based on External Laser Interferometer: Design, Control and Experimental Validation. Sensors 2017, 17, 838.
Author Response
Dear Reviewer:
Thank you for your letter and for the reviewer’s comments concerning our manuscript entitled “Research on A New Type of Rigid-flexible Coupling 3-DOF Micro-positioning Platform” (micromachines-972821). Those comments are all valuable and very helpful for revising and improving our paper, as well as the important guiding significance to our researches. We have studied comments carefully and have made correction which we hope meet with approval. Revised portion are marked in yellow in the paper. The main corrections in the paper and the responds to the reviewer’s comments are as follows:
Point 1: There are 'flexible drive mechanism' and 'flexible driving mechanism' used within the whole paper. It will be better if there is only one consistent expression used. 

Response 1: In the revised manuscript, line 18, the statements of “flexible driving mechanism” are corrected as “flexible drive mechanism”.
Point 2: In section 2.3.2 Dynamic analysis, there is only modal analysis, so it should be changed as Modal analysis.
Response 2: In the revised manuscript, line 194, the statements of “Dynamic analysis” are corrected as “Modal analysis”
Point 3: In section 2.3.2, the lowest natural frequency is about 1797 Hz without considering the 'rigid platform' part. And in section 3.2.3, the frequency test shows that with 50 Hz frequency, the amplitude changes. As this test is conducted with the rigid platform (I assume), so in section 2.3.2, it will be better to include the results of modal analysis with the rigid platform.
Response 3: In section 2.3.2, the rigid platform is included in the experimental tests, which is consistent with the review’s opinion. According to the reviewer's suggestion, the rigid platform is also considered in the modal analysis of section 2.2.3 of the revised manuscript. Simulation results are shown in Figure 6. Meanwhile, “The first four natural frequencies are 1797.03 Hz, 1799.51 Hz, 1805.63 Hz and 6796.3 Hz, respectively.” is corrected as “The first four natural frequencies are 4508.21 Hz, 4590.59 Hz, 4890.59 Hz and 6020.79 Hz, respectively.” in the manuscript (revised in lines 199-200 on page 6).
Point 4: From figure 10-13, when the amplitude is 40V, the displacement is 0-7,0-11 in figure 10; when the amplitude is 90V, the displacement is 25-50, 80-110, 65-110 in figure 11; The same issues also happen in figure 12-13. The authors should carefully check the corresponding displacements under different amplitudes of voltages and then modify these figures.
Response 4: When the input voltage amplitude is 40V and 90V, all output displacements have been carefully checked and the results are correct. However, what we want to explain is that the distance between the moving piece and the fixed piece of the capacitive sensor is manually adjusted and set in advance during the test, which leads to that the centre point of the output displacement corresponding to the input of different voltage amplitudes is not fixed. It is same as the reviewer’s opinion (when the voltage is 90V, the output displacement range is 25-50, 80-110, 65-110). In order to facilitate the analysis of the output displacement, data processing should be performed on it, that is, unified adjustment is based on 0. In the original manuscript, the output displacement of Figure 10 is after data processing, but the output displacement of Figure 11-13 is not processed. This is the author's sloppy, which has been corrected in the revised manuscript (revised on pages 11-14 ).
Point 5: It would be better if the authors could summarize the characteristics of the developed micro positioning platform in a table and add also some comparison with some other similar platform already developed in other references.
Response 5: We summarize the characteristics of the developed micro-positioning platform and also compares with other similar platforms (revised in lines 114-129 on pages 3-4).
The movement of this traditional flexure hinge device is realized by the deformation of the flexure hinge, which has advantages of frictionless drive and strong anti-interference ability. Tang et al. [32] designed a micro-positioning platform with fast response and maximum crosstalk less than 1.2%. Liu et al. [33] designed a three-axis precision positioning device using a low-interference motion piezoelectric actuator. However, the flexible structure must have greater flexibility in two decomposition directions at the same time when this type of device achieves force decomposition through a flexible hinge. For example, when Z-shaped hinge structure is used as force decomposition unit, it is required to have greater flexibility in both decomposition directions. Therefore, when the external force is added, it is easy to cause hinge deformation and produce disturbance error [34]. The micro-positioning platform proposed in this study is composed of two parts: a flexible drive mechanism and a rigid platform. The flexible drive mechanism and the rigid platform are connected by inclined surface contact to form a rigid kinematic pair. The rigid kinematic pair can not only play the role of force decomposition, but also ensure that the device has a large output rigidity, which is beneficial to improve the anti-interference ability and the load-bearing capacity of the device. Meanwhile, the piezoelectric ceramic driver is protected by the reverse friction self-locking feature, which is especially suitable for precise positioning of large-quality workpieces.
Point 6: There is some recent literature which the authors can include in the references to make the paper cover more recent development of micro-positioning platform, for example:
Li, P.-Z., etc. Piezoelectric Actuated Phase Shifter Based on External Laser Interferometer: Design, Control and Experimental Validation. Sensors 2017, 17, 838.
Response 6: We have added five additional references, as follows:
- Li, P.; Wang, X.; Sui, Y.; Zhang, D.; Wang, D.; Dong, L.; Ni, M. Piezoelectric Actuated Phase Shifter Based on External Laser Interferometer: Design, Control and Experimental Validation. Sensors. 2017, 17, 838.
- Li, J.; Liu, H.; Zhao, H. A Compact 2-DOF Piezoelectric-Driven Platform Based on “Z-Shaped” Flexure Hinges. Micromachines. 2017, 8, 245
- Tang, C.; Zhang, M.; Cao, G. Design and testing of a novel flexure-based 3-degree-of-freedom elliptical micro/nano-positioning motion stage. Advances in Mechanical Engineering. 2017, 9, 1-10.
- Liu, Y.; Li, B. A 3-axis precision positioning device using PZT actuators with low interference motions. Precision Engineering. 2016, 44, 118-128.
- Xie, Y.; Li, Y.; Cheung, C.; Zhu, Z.; Chen, X. Design and analysis of a novel compact XYZ parallel precision positioning stage. Microsystem Technologies. 2020. DOI: 10.1007/s00542-020-04968-6
The changes are as follows:
Lines 83-85, “Li et al. [30] use three pairs of parallel four-bar design of flexure hinge mechanism of a 3-DOF platform for micro-positioning platform. Li et al. [31] designed a micro-positioning platform with high positioning resolution.” was added
Lines 115-122, “Tang et al. [32] designed a micro-positioning platform with fast response and maximum crosstalk less than 1.2%. Liu et al. [33] designed a three-axis precision positioning device using a low-interference motion piezoelectric actuator. However, the flexible structure must have greater flexibility in two decomposition directions at the same time. For example, when Z-shaped hinge structure is used as force decomposition unit, it is required to have greater flexibility in both decomposition directions. Therefore, when the external force is added, it is easy to cause hinge deformation and produce disturbance error [34].” was added.

Reviewer 2 Report
The paper is well structured and written.
Authors used a proper scientific and systematic approach in presenting their micro positioning system.
Their conclusions are supported by FEM, simulation and experimental analyses.
Would be interesting to explore how the piezoelectric actuators would affect the motion precision of the rigid platform.
Author Response
Dear Reviewer:
Thank you for your letter and for the reviewer’s comments concerning our manuscript entitled “Research on A New Type of Rigid-flexible Coupling 3-DOF Micro-positioning Platform” (micromachines-972821). Thank you very much for your approval.
Round 2
Reviewer 1 Report
The authors have replied to the comments and modified the manuscript. It is now suitable for publication.